# Foundation models for high-energy physics

**Anna Hallin**[1⋆]

**1** Institute for Experimental Physics, University of Hamburg, Luruper Chaussee 149, 22761 Hamburg, Germany

⋆ anna.hallin@uni-hamburg.de

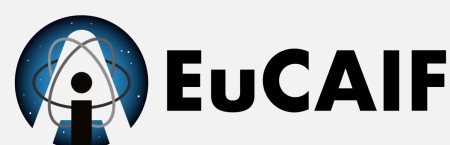

*The 2nd European AI for Fundamental Physics Conference (EuCAIFCon2025) Cagliari, Sardinia, 16-20 June 2025*

## Abstract

**The rise of foundation models – large, pretrained machine learning models that can be finetuned to a variety of tasks – has revolutionized the fields of natural language processing and computer vision. In high-energy physics, the question of whether these models can be implemented directly in physics research, or even built from scratch, tailored for particle physics data, has generated an increasing amount of attention. This review, which is the first on the topic of foundation models in high-energy physics, summarizes and discusses the research that has been published in the field so far.**

## 1 Introduction

The term *Foundation models* was invented by researchers at the Stanford Institute for Human-Centered Artificial Intelligence [1], to discuss the implications of the rise of large pretrained models such as BERT [2], GPT-3 [3], and CLIP [4]. Their definition of foundation models is as follows:

> *A foundation model is any model that is trained on broad data (generally using self-supervision at scale) that can be adapted (e.g., fine-tuned) to a wide range of downstream tasks (...)*

In essence, this definition describes transfer learning, which per se is nothing new [5]. However, what was new at the time was the scale of the models, the size of the datasets they were trained on, and the amount of compute available for training. GPT-3, for example, has 175 billion model parameters and was trained for a total of 300 billion tokens, requiring $\mathcal{O}(10^{23})$ flops of compute for training. With increasing scale came not only increasing capabilities, but abilities that the model was not trained for or expected to acquire were also seen to emerge [6]. Having access to large amounts of data, computing resources, as well as machine learning expertise, there has been a growing interest within the high-energy physics (HEP) community for

these types of models. However, views differ on what capabilities a model needs to possess in order to qualify as a foundation model. In this review, the definition above will be interpreted generously, such that a foundation model is defined as being any machine learning model that fulfills the following:

1. It has been pretrained on a large amount of data, with the explicit intention of creating a rich latent representation that serves as a foundation for other tasks;

2. It uses this latent representation to finetune the model to a different downstream task (be it the same type of task on a different dataset, or a different task altogether);

3. The final performance on the downstream task improves when the latent representation is used, compared to if the model is initialized without this representation.

Additional criteria such as self-supervised learning[1], few- or zero-shot capabilities [7], multi-modality [8] or criteria for the number or diversity of downstream tasks, will not be required here. Note that while large language models (LLMs) and vision models are what made foundation models famous, the concept is not restricted to this data type.

The structure of the paper is as follows. Section 2 provides examples of different approaches to foundation models in HEP, and outlines the potential benefits the field could gain from their implementation. Section 3 reviews one foundation model in detail, whereas section 4 compares several different models published in the past few years. Brief conclusions are presented in section 5.

## 2 Motivation

The current approaches to utilizing foundation models in HEP can be divided into three categories[2]:

1. Existing foundation models like LLMs can be employed directly to assist in physics research. Examples include agentic systems capable of tuning particle accelerators [10] or searching for anomalies in particle physics data [11].

2. While LLMs excel at text, they are not necessarily experts at mathematical reasoning [12]. Efforts at alleviating this include using a distinct embedding for numbers [13], enabling the model to understand that numbers are different entities than text and that they behave differently. Another approach has been to represent mathematical expressions as sequences, and to treat equation solving and integration as a translation task [14].

3. Finally, one does not need to restrict oneself to working with existing language and vision models or the concepts they build upon, but can attempt to build foundation models for HEP from scratch.

The remainder of this review will mainly concern works based on the third approach, with a focus on collider physics. Collider experiments at the LHC like ATLAS [15] and CMS [16] collect vast amounts of highly complex and structured data. The data, coming from different subdetectors, is inherently multimodal. In addition, simulated (and therefore, labeled) datasets of high quality exist. As foundation models need large amounts of data for pretraining, this setting provides an excellent opportunity to develop and implement such models, investigating

---

[1]In contrast to many other fields, particle physics, and collider physics in particular, has access to high-quality simulations (i.e. labeled datasets), which enables supervised learning at scale.

[2]For a wider perspective on foundation models as part of physics-specific large-scale AI models, see [9].

their capabilities in capturing intricacies and correlations in the data that we would perhaps not be able to untangle otherwise. Furthermore, physicists perform a wide range of downstream tasks using this data, including event classification, object tagging, anomaly detection, clustering, generation and regression. Machine learning methods are already being implemented in several of the experimental stages, from triggering [17, 18] to analysis [19, 20], meaning that expertise and experience already exist in the field. Having access to a rich latent representation of the data could also boost the achievable performance on small datasets, relevant in the context of rare processes where the amount of available simulation is limited due to narrow selection criteria.

Beyond possible gains in extracting and analyzing data, the implementation of foundation models could also result in higher efficiency, saving both computational and human resources. With the HL-LHC [21], the high luminosity phase of the LHC, around the corner, the amount of required computing resources is expected to skyrocket [22,23]. While the pretraining phase of foundation models is resource-intensive, subsequent finetunings would require less resources compared to training each downstream task from scratch.

## 3   An in-depth foundation model example

The first cross-task foundation model published for collider physics, capable of both generation and classification of particle jets, is called OmniJet-$\alpha$ [24]. The pretraining target is generation, with the idea being that a model that knows how to generate jets should understand something about jets in general, and thus be suitable for downstream tasks such as classification. OmniJet-$\alpha$ was developed using the simulated JetClass dataset [25], originally released together with the Particle Transformer [26]. The low-level features $p_{\mathrm{T}}, \Delta\eta$ and $\Delta\phi$ are tokenized via a Vector Quantized Variational Autoencoder (VQ-VAE) [27–30], leading to jets being represented as sequences of integers rather than sets of constituents with multiple features per constituent. The model is based on the transformer architechture [31], essentially a smaller version of the original GPT-1 model [32] but without positional encoding, and is trained in a self-supervised fashion with next-token prediction as target. This means that the model learns the probability $p_j$ of token $x_j$ to follow a sequence of previous tokens: $p_j = p(x_j|x_{j-1},...,x_0)$. This makes generation straight-forward: the model is provided with a start token (a special token not representing a particle) and then samples its learned distribution to autoregressively generate a full jet. At the end, the VQ-VAE is used to decode the tokens back to physics space. The original paper showed good agreement between "real" and generated jets, and demonstrated that the pretrained model outperformed a model with random initializations on the classification task, in particular for very small datasets.

Since OmniJet-$\alpha$ is not dependent on labeled data for pretraining, it is able to train directly on data. This was done using the Aspen Open Jets dataset [33], a derivate jet dataset extracted from CMS Open Data [34,35]. While not labeled, this dataset is expected to mainly contain jets originating from light quarks and gluons. It was shown that pretraining on these jets proved helpful for the downstream task of generating hadronically decaying top jets (with a finetuning training sample from JetClass), in particular for quantities like the n-subjettiness [36,37] that is difficult to model [38].

A benefit of the tokenized approach is that the architecture can be immediately re-used for other data types. The model itself only requires sequences of integers, which means that as long as the data can be tokenized, the framework does not need to be adapted to any other data format. The capability of the model to switch domains was tested using pointcloud calorimeter showers [39]. Since the jet domain and the calorimeter shower domain are assumed to have no cross-over capabilities, the weights from the jet model were not re-used

in this study. It was shown that although slower than other generative models for calorimeters (see e.g. [40] for examples), the model was indeed capable of generating photon showers with performance on par with two models dedicated to calorimeter shower generation [41, 42].

Beyond jets and calorimeters, OmniJet-$\alpha$ has also been applied to $\tau$ physics [43], including tasks such as $\tau$ identification, kinematic reconstruction, and determination of its decay mode.

## 4  Comparing existing foundation models

This section includes models that fit the foundation model criteria as outlined in the introduction, and focus on collider physics including particle jets[3]. Some of these models do not refer to themselves as foundation models, but since they fit the criteria they are included nonetheless to show what different approaches have been investigated so far[4].

- Particle Transformer (ParT) [26] (Feb 2022) pretrains on a supervised classification task, utilizing jet constituent features: kinematic, particle ID, trajectory discplacement and interaction features (the latter inspired by the jet clustering history [46]). Downstream tasks include finetuning to classification on different datasets.

- Masked Particle Modeling [29, 47] (Jan 2024), inspired by BERT [2], is pretrained on the self-supervised task of predicting masked out portions of the jets. The initial version used jet constituent features (kinematics only), with the targets being tokenized, whereas the second version used continuous features all-over, expanding the feature set to include kinematics, particle ID and trajectory displacement. The model was finetuned on classification (fully and weakly supervised, respectively).

- OmniJet-$\alpha$ [24] (Mar 2024), described in detail above, is inspired by GPT-1 [32] and pretrained on a self-supervised next-token prediction task, using tokenized jet constituent kinematic features. Downstream tasks include generation and supervised classification.

- Re-simulation-based self-supervised learning (RS3L) [48] (Mar 2024) pretrains using contrastive learning, aiming to group augmentations of the same simulated process and separate them from the other events. The augmentations are created by fixing the hard process and then re-running the showering, hadronization and detector response steps of the simulation[5]. Features used are jet constituent kinematic features, particle ID and trajectory displacement features. The model is finetuned to supervised classification.

- OmniLearn [52, 53] (Apr 2024) utilizes a hybrid pretraining task, combining generation and supervised classification. The model is trained on both jet and constituent kinematics, as well as particle ID. A dropout function for particle ID allows finetuning on datasets lacking this feature without re-training the backbone. Downstream tasks include generalizing to classification across jet types, detectors and collision systems; conditional generation, reweighting, unfolding and anomaly detection.

- L-GATr [54] (Nov 2024), while not claiming to be a foundation model, fits the criteria outlined in the introduction as it demonstrates pretraining and finetuning. The model is based on a Lorentz-equivariant architecture with mechanisms for symmetry breaking,

---

[3]Foundation models for collider physics *not* including particle jets include applications to nuclear physics [44, 45].

[4]Works that use similar techniques as the ones described in the following but do not perform any finetuning or indicate any foundation model intent, will not be included here.

[5]See e.g. [49] for an introduction to the first three simulation steps at hadron colliders, and [50, 51] for a examples of detector response simulation methods.

and particles are represented using multivectors in a geometric algebra. The model is pretrained on supervised classification using jet constituent kinematic features, and finetuned to supervised classification on other datasets. This model is the current state-of-the-art for supervised classification.

- Jet-based joint embedding predictive architecture (J-JEPA) [55] (Dec 2024) builds on the JEPA approach [56,57], making predictions in the latent space rather than in feature space. Large-radius jets are re-clustered into smaller subjets, and the pretraining task is to predict masked out subjets in the representation space. After pretraining, the model is finetuned to supervised classification.

- Bumblebee [58] (Dec 2024) aims to build a foundation model for whole events, rather than single jets, using only high-level features (i.e., treating jets as single objects rather than collections of objects). Simulated events both at generator-level and reconstruction-level are used for pretraining, where the task is to predict masked out particles. In the first half of the training, random particles are masked out, and in the second half, either all generator-level or all reconstruction-level particles are masked out. Downstream tasks include top quark reconstruction and two supervised classification tasks.

- Pretrained event classification model for high energy physics analysis [59] (Dec 2024) pretrains on entire events in a supervised fashion either on multi-class classification or so-called multi-label learning tasks. The downstream task is supervised binary event level classification, including new event types that were not seen during pretraining.

- HEP-JEPA [60] (Feb 2025) is very similar to J-JEPA, however, instead of masking out subjets HEP-JEPA forms patches – groups of particles – inside the jet, which are represented by an embedding. Part of these embeddings are then masked, with the pretraining task being to fill in the masked-out patches. The input data consist of kinematic features for the jet constituents, and interaction features calculated between patches. Downstream tasks include multi-class and binary classification.

The models above differ along several axes – three of them will be selected for this comparison. The most obvious one is the **pretraining task**. The two main types of pretraining tasks are "fill in the blanks" – either via masked prediction (MPM, J-JEPA, Bumblebee, HEP-JEPA) or next-token prediction (OmniJet-$\alpha$) – and classification (ParT, OmniLearn, L-GATr, Pretrained event classification). RS3L is the only model using contrastive learning, while OmniLearn and Pretrained event classification stand out using hybrid approaches, combining classification with generation and regression respectively. The reasons for choosing these particular pretraining tasks are varied. In the case of ParT and L-GATr it is straightforward: the aim is to boost the classification performance without attempting any other downstream tasks. This makes the choice of classification as pretraining task quite obvious: at the time of their release, these models outperformed all other models on the selected classification tasks. In the case of OmniLearn, the authors state that using the same type of task for pretraining as you want to perform in the downstream tasks increases the effective size of the training set for the downstream task. According to the authors, this expansion of the training data is likely what lies behind the usefulness of foundation models. Hence the choice of a combined generative/classification pretraining task. Some models, like OmniJet-$\alpha$ and Bumblebee, employ pretraining tasks that feed directly into one of their downstream tasks, generation and reconstruction respectively. This makes the pretraining immediately useful in itself, requiring no further finetuning. The goal behind the pretraining task in RS3L is to reach domain completeness – to cover as much of the stochastic space inherent to the simulation tools as possible. Although there seems to be some truth to the statement that aligning the pretraining

task with the desired downstream tasks could lead to an increase in performance, it is not yet known whether tailoring a foundation model precisely according to the exact (limited number of) tasks you want it to perform, could restrict the achievable performance on new tasks or emergent behavior at scale. It will definitely limit the re-usability of the architecture across subdomains, a property that may or may not be desirable.

Another axis concerns the level of **supervision** and whether **simulations** are necessary for pretraining. Many of the models (ParT, OmniLearn, L-GATr, and Pretrained event classification in the multi-class version) require labels, and some of the models that in principle do not require labels still rely on simulation (RS3L, Bumblebee) and can thus not be pretrained directly on data. The exceptions that do not require neither labels nor simulation for pretraining are OmniJet-$\alpha$, MPM and the two JEPA models. High energy physics differs from many other fields in that we have access to high-fidelity simulations. However, simulation is not completely free, in particular not for rare processes, and it is not perfect. It is not yet known whether pretraining on simulation rather than data, in cases where labels are not required, harms the performance of the model. What we do know is that a model that is not dependent on simulation is still able to train on it, whereas a model dependent on simulation closes the door to including data as part of its training.

The amount of **required physics information** differ between the models. Most models fix the input type or selection of features, apart from OmniLearn which allows particle ID to be dropped, and OmniJet-$\alpha$ which can use any type of data as long as it can be tokenized. When it comes to using low level (constituent) or high level (jet) features, most jet-specific models use low level features while the event-level models use high-level features. The two JEPA implementations land somewhere in the middle, using either subjets (J-JEPA) or patches (HEP-JEPA). Already with ParT it was clear that adding more physics information, as long as it is relevant for the task, helps the model. L-GATr takes this the furthest, encoding Lorentz equivariance in the model architecture itself.

## 5  Conclusion

Foundation models for high energy physics are highly interesting as they may help us reach better physics results – whether this comes from a stronger performance overall, or more efficient use of resources. The field as a whole has the experience, the expertise, the data volumes and the computing resources needed. Presently, several different pretraining strategies have been explored, and we are likely to see new ideas in the coming years. The development of these types of models, however, requires not only new ideas and approaches developed by individual research groups. The resource requirements are potentially huge, which calls for increased cooperation, and exchange of ideas and experiences with groups that work on foundation models for other fields, including research into scaling and emergent behavior.

## Acknowledgements

I wish to extend my gratitude to EuCAIF and the local organizing committee for the excellent organization of this conference, and for giving me the opportunity to present this work. In addition, I am grateful to Tobias Golling and Lukas Heinrich for organizing the working group discussion session on foundation models, and to Joschka Birk and Gregor Kasieczka for valuable comments on this manuscript. This work was supported by the DFG under the German Excellence Initiative – EXC 2121 Quantum Universe – 390833306 and under PUNCH4NFDI – project number 460248186.

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
