# Peer review of "Foundation models for high-energy physics"

_SciPost Physics Proceedings_

## Round 1 · Referee Report · Tobias Golling (Referee 1) · 2025-10-29

Report

Excellent summary of the state of foundation models in our community.

Requested changes

Please update this reference please:
M. Leigh, S. Klein, F. Charton, T. Golling, L. Heinrich, M. Kagan, I. Ochoa, M. Osadchy, “Is Tokenization Needed for Masked Particle Modelling?,” 2025 Mach. Learn.: Sci. Technol. 6 025075 [2409.12589].
See https://iopscience.iop.org/article/10.1088/2632-2153/addb98

For ATLAS references please use ATLAS Collaboration for the author, same as done for CMS (see e.g. Ref. 18 and 20)

Refs 17-20 are a bit arbitrary. I don't have a good suggestion how to fix it. Either add many more references to give a representative view or focus on summary papers or remove them altogether as they are not relevant in the end.

Suggest to elaborate on the possible modalities in HEP (such as Standard high-level reco objects, Pflow objects, Tracks and clusters, Raw data (hits…), various (latents) truth modalities, trigger objects as well as language).

In Section 2 the case is made that the focus will rest on bullet three. This is well justified as the main work in the community is on this bullet. This should be clarified and maybe the order adjusted according to the relevance in our community, i.e. start with bullet 3.

I find Section 3 problematic. I understand that the author is an expert on this particular model, however as this document claims to be a review paper on the state of the art (see abstract), I advocate against using it as a platform to highlight the own work. Maybe this is just my personal preference. But I worry it sends the wrong message to the community. My suggestion is to drop this section. Some of the details could be moved to Section 4 but care should be taken to keep a healthy balance between the various methods and approaches.

A final (personal) request: it would be very useful for the community to arrange and compare the methods visually along the various axes which are mentioned in the text. I know that the information is already in the current text. Would be nice to have this - if it is not too much work. Such a sketch could become a reference in our community to show the most relevant differences (regarding pre-training techniques, supervision, physics encoding, input features, downstream tasks, etc.) - as was done to some extent in the talk.

Last sentence in the conclusion: I am wondering if you wish to reference EuCAIF WG1 directly as an opportunity here?

Recommendation

Ask for minor revision

---

## Editorial Decision

awaiting_resubmission